# Diagnostic Accuracy of LDBIO-Toxo II IgG and IgM Western Blot in Suspected Seroconversion in Pregnancy: A Multicentre Study

**DOI:** 10.3390/pathogens11060665

**Published:** 2022-06-08

**Authors:** Valeria Meroni, Francesca Genco, Luigia Scudeller, Marie-Pierre Brenier-Pinchart, Hélène Fricker-Hidalgo, Coralie L’Ollivier, Luc Paris, Hervé Pelloux

**Affiliations:** 1Department of Molecular Medicine, University of Pavia, 27100 Pavia, Italy; v.meroni@smatteo.pv.it; 2Microbiology and Virology Unit, Fondazione IRCCS San Matteo Hospital Foundation, 27100 Pavia, Italy; francesca.genco@asst-lodi.it; 3UOC Innovation and Research, IRCCS University Hospital Sant’Orsola, 40138 Bologna, Italy; luigia.scudeller@aosp.bo.it; 4Parasitology-Mycology, University Hospital and Grenoble-Alpes University, 38043 Grenoble, France; mppinchart@chu-grenoble.fr (M.-P.B.-P.); hfricker-hidalgo@chu-grenoble.fr (H.F.-H.); hpelloux@chu-grenoble.fr (H.P.); 5Aix Marseille Univ, IRD, AP-HM, SSA, VITROME, IHU Méditerranée, 13005 Marseille, France; coralie.lollivier@ap-hm.fr; 6APHP-Sorbonne Université, Parasitology-Mycology, Pitié-Salpêtrière Hospital, 75013 Paris, France

**Keywords:** toxoplasmosis, seroconversion, pregnancy, IgG, IgM, western blot

## Abstract

The high sensitivity of the automated tests used for *Toxoplasma gondii* serology can yield false-positive IgM results due to aspecific reactions. On the other hand, specific therapy can delay IgG production and, therefore, the diagnosis of seroconversion. There is a need for confirmation tests to early detect seroconversions during pregnancy. We conducted a multicentre study to evaluate the diagnostic accuracy of the Toxo II IgG and a new, not yet commercialised Toxo II IgM western blot (WB) (LDBio diagnostics Lyon France) on 229 sera corresponding to 93 patients with seroconversions and 158 sera corresponding to 68 patients with nonspecific IgM. Sensitivity was 97.8% for IgM WB and 98.9% for IgG WB. Specificity was 89.7% and 100%, respectively. The concordance between IgM and IgG Toxo WB with the final diagnosis was very good, K = 0.89 and K = 0.99, respectively. In 5 cases (5.4%), the appearance of IgM, and in 55 cases (59.1%), the appearance of IgG was recorded by WB earlier than by traditional tests. In 10 cases (10.8%), IgM was detected after the traditional tests and in 2 cases (2.2%) for IgG. The association of IgG and IgM WB on the same sample not only detected all seroconversions but also correctly identified most of the false-positive results.

## 1. Introduction

Toxoplasmosis is a ubiquitous protozoan infection due to the Apicomplexa *Toxoplasma gondii*. Infection is usually asymptomatic or mildly symptomatic in immunocompetent patients, with spontaneous recovery, but can be life-threatening in immunocompromised patients (HIV positive, transplanted patients, patients with haematological disorders) and cause severe disorders (abortion, stillbirth, hydrocephalus, chorioretinitis) in congenitally infected foetuses from mothers with acute infection during pregnancy [1,2]. Therapy with spiramycin as a first-line and/or pyrimethamine plus sulfadiazine as an alternative regimen seems to be effective in reducing the burden and the severity of congenital toxoplasmosis if given in the first weeks after the infection [3,4]. To this end, several countries in Europe have adopted a serological screening programme in order to define the immunological status of pregnant women [5]. Routine testing relies on the determination of anti-*Toxoplasma* specific IgG and IgM antibodies in serum, on monthly follow-up visits and counselling for hygienic–prophylactic measures in seronegative women, and if necessary, treatment of women who seroconverted [3,5,6].

Automated tests are usually employed for the determination of specific IgG and IgM antibodies, but despite vast improvements in test performance in recent years, equivocal results with low levels of IgM without IgG in previously seronegative women are still frequently observed in clinical practice due to a lack of standardisation and high sensitivity of most of the marketed tests. They are thus sometimes difficult to interpret, even for medical parasitologists. In a pregnant woman with a suspected recent *Toxoplasma gondii* infection (IgG negative and IgM positive), the differential diagnosis between seroconversion and false IgM positivity is often difficult, as the infection is in most cases asymptomatic, and it is necessary to wait until specific anti-*Toxoplasma* IgG antibodies are measurable [7,8,9,10]. Furthermore, treatment with spiramycin, and even more with sulfadiazine and pyrimethamine, prescribed once IgM is detected, can delay IgG production and mask seroconversion [11].

A diagnosis of toxoplasmosis in pregnancy requires treatment until delivery, amniocentesis to diagnose infection in the foetus and serological and clinical follow-up of the newborn.

This indicates the need for an early and precise diagnosis during pregnancy in order to treat only true seroconversions as soon as possible.

The diagnostic accuracy of LDBio Toxo II IgG^®^ western blot (Toxo II IgG) as a second-line, confirmatory test is already established. The test has received the CE mark, and it is recognised as a very good alternative to the Sabin–Feldman dye test, which is still considered the gold standard [12,13,14,15,16,17,18,19].

The time to anti-*Toxoplasma gondii* IgG detection during infection in pregnant women was significantly shortened by the use of Toxo II IgG WB [7,13]. Toxo II IgG WB early detects a toxoplasmic seroconversion that can either reduce the time to therapeutic intervention or avoid unnecessary and expensive serological follow-up. In our study, we aimed to evaluate the diagnostic accuracy (sensitivity and specificity) of the combined Toxo II IgG and, for the first time, a new test, unmarketed and not yet commercialised, the Toxo II IgM western blot (LDbio Diagnostics, Lyon France) in the diagnosis of seroconversion in pregnant women.

## 2. Materials and Methods

### 2.1. Serum Samples

We performed a retrospective multicentre study involving four referral centres for the diagnosis of toxoplasmosis (Microbiology, Virology Unit IRCCS Policlinico San Matteo Pavia, Italy; Parasitology-Mycology, University Hospital and Grenoble-Alpes University, Grenoble, France; Pitié-Salpêtrière Hospital, AP-HP Sorbonne University, Parasitology-Mycology, Paris, France, Aix Marseille University, IRD, AP-HM, SSA, VITROME, IHU Méditerranée, Marseille, France) on the leftovers of laboratory samples.

For seroconversion patients, at least two samples were collected. The first sample had to be either the last sample with negative results, whenever available, or the first sample with IgM (IgG negative or positive). In all the first samples, the IgG titer with traditional tests was very low, often negative, or in the grey zone, according to the threshold recommended by each manufacturer.

For false-positive IgM, at least one sample was kept for the study, but whenever available, further samples without the appearance of IgG were also included. The clinical status of the patients from whom each sample was drawn was determined by each centre according to the combination of tests used in each laboratory (Table 1); the presence or absence of IgG on sera collected later confirmed or invalidated the seroconversion and the false-positive IgM.

Inclusion criteria for the samples were full anonymisation, samples stored frozen at no more than −15 °C and available volume of at least 400 μL (200 μL for testing and 200 μL in storage for retesting in case of discrepancies). Patient samples were selected from those already in local registries at participating centres with a signed informed consent to the use of the residual samples. Centres had to ensure that all the clinical and laboratory information needed for diagnosis was available. All samples present in the existing database of each centre analysed in the previous 6 years [20] were included. After selection, the samples were shipped frozen (dry ice) to LDBio Diagnostics (LDBio, Lyon, France) for sample processing. Two simple Excel databases were built before sample selection and sent to each participant centre, one for the diagnostic tests of each sample (sent to clinical centres) and one for the WB test (sent to LDBio, together with the serum samples and including only sample code). All coded samples were analysed blindly at LDBio, Lyon, France. Populated databases were sent to the coordinating centre in Pavia for coupling of the reference test and WB Toxo test results and for statistical analyses (Clinical Epidemiological Unit, Fondazione IRCCS Policlinico San Matteo Pavia Italy).

### 2.2. Methods

LDBio performed two tests on all the samples: TOXO II IgG was run following the corresponding instructions for the use (IFU) of the commercial kit, and Toxo II IgM, for which no IFU is available, as the test is not commercialised, was performed under the same conditions as Toxo II IgG (sample volume = 10 µL, incubation times = 90′–60′–60′) using the commercial LDBio anti-IgM conjugate. The cutoff for IgG WB was set following the criteria of the IFU by the presence of three of five specific bands (30, 31, 33, 40 and 45 kDa), including the 30 kDa band. The antigen is a *Toxoplasma* lysate obtained in vivo. The bands have been defined experimentally as being found in patients. Their identification was made by comparison with molecular weight markers and, therefore, corresponds to an approximate molecular weight. The P30 protein seems to be SAG1, the major *T. gondii* surface protein and the most immunogenic constituent of tachyzoïtes. The cutoff of Toxo II IgM WB was defined by the presence of two bands in the specific area (between 30 and 40 kDa) and including the P30 (the profile of Toxo II IgM is different from the profile of Toxo II IgG: the specific bands are P30, P31, P33, P38 and P40). The presence of a single P30 or very pale bands was also recorded and interpreted as equivocal. The criteria of the IFU requiring three bands for IgG did not lower sensitivity compared with using two bands [12]. This also reduces the risk of misinterpretation due to positivity to nonspecific bands close to the specific reading area. For IgM WB, a cutoff at two bands was suggested by the manufacturer prior to the study as per nonpublished data.

Toxo II IgG and Toxo II IgM were performed with the help of a Dynablot plus instrument (Dynex, Buštěhrad, Czech Republic) by an operator blinded to reference test results within LDBio facilities.

### 2.3. Statistical Analysis

To validate the test, two different analyses were performed: concordance (Cohen’s kappa) with final diagnosis (seroconversion v. false-positive) and diagnostic accuracy (sensitivity, clinical specificity, proportion of correct results, with the corresponding binomial 95% confidence intervals). With a sample size of 100 patients with proven seroconversion and 100 with nonspecific IgM, we estimated >95% power to identify both sensitivity and specificity of 90% (v. alternative hypothesis of 75%, with alpha error < 0.05). The unit of analysis was the patient (not the samples) to avoid over/under estimation of performances due to repeated samples of the same patient. At the same time, an analysis at the sample level was performed to determine which technique detected IgG and IgM first.

We performed two sensitivity analyses for equivocal IgM WB results, considering them once as negative and once as positive, even if, for clinical purposes, it might be better to consider the equivocal results as positive, as they could be an earlier marker for seroconversion. Finally, the frequency of antigenic bands for positive samples was calculated. Stata computer software version 16.0 (Stata Corporation, 4905 Lakeway Drive, College Station, TX 77845, USA) was used for the statistical analyses.

## 3. Results

Three hundred and eighty-seven sera were collected: 229 corresponding to 93 toxoplasmic seroconversions (2 to 4 sera/patient) and 158 sera corresponding to 68 patients with cross-reactions and/or nonspecific IgM obtained with the automated tests and/or IgM ISAGA Tests (1 to 7 sera/patient) (Table 1). In the case of seroconversion, the delay between two sera was generally between 2 and 3 weeks (less than 1 month), as recommended by the French Toxoplasmosis National Reference Center. Toxo II IgM WB was positive for 91/93 seroconversions and equivocal in 2/93. It was also positive in 7 (14 samples) and equivocal in 19 (27.2%) (43 samples—one patient had both equivocal and positive results and was, therefore, classified as positive) out of 68 false-positive IgM cases (Table 2). Toxo II IgG WB was positive in 92/93 true seroconversions and correctly identified as negative in all 68 patients with nonspecific IgM. The sensitivity of Toxo II IgM WB with equivocal results considered negative and positive was respectively 97.8% (95CI 91.7–99.6%) and 100% (95CI 95.0–100%). The sensitivity of Toxo II IgG WB was 98.9% (95CI 93.3–99.9%). The specificity of Toxo II IgM WB with equivocal results considered negative and positive was respectively 89.7% (95CI 79.3–95.4%) and 61.8% (95CI 49.1–73.1%). The specificity of Toxo II IgG WB was 100% (95CI 93.3–100%). The individual comparison between WB and each traditional test can be found in the S1 (IgM) and S2 (IgG) files. For IgM and IgG WB, the concordance with the final diagnosis was very good: K = 0.89 and K = 0.99, respectively. (Table 2) Among the discordant results (WB positive and traditional tests negative, or WB negative and traditional tests positive), there were 5 cases (5.4%) in which the appearance of IgM and 55 cases (58%) in which the appearance of IgG in seroconversion was recorded by WB before the traditional tests, and 10 cases (10.8%) in IgM and 2 cases (2.2%) in IgG in which traditional tests were positive first. Looking at the most antigenic bands, P30 was positive in 203/229 (88.6%) and P40 in 177/229 (77.3%) of the samples from seroconverted patients. P30 was positive in at least one sample for all positive patients and P40 in all but eight (88.4%), as the appearance of specific bands in these patients was only delayed.

The interpretation of the results was easy and reproducible. Some examples of WB strip results are shown in Figure 1 for seroconversion from the first negative sample to the positivity of specific IgG and IgM through the appearance of IgM 30–40 kDa. The false-positive results are reported in Figure 2; these strips showed a different antigenic response that remained constant over time, characterised by the absence of three bands for IgG and two bands for IgM on the P30–40 proteins.

## 4. Conclusions

In this multicentre study, we evaluated for the first time the diagnostic efficacy of Toxo II IgM WB in a selected population of patients positive for anti-*Toxoplasma* IgM and negative for anti-*Toxoplasma* IgG. Therefore, the performances of the tests in this study do not reflect their actual performances in the field, but only in this specific population, in which diagnosis is difficult due to the puzzling serology results. The aim here was to answer the question: is it real seroconversion or false IgM positivity?

The Toxo II WB IgM not only detected almost all seroconversions with a sensitivity of 97.8% but also well discriminated the false-positive results (Table 2) with a specificity of 89.7% on otherwise false-positive samples with screening techniques, showing superior specificity to all other techniques in this setting (Appendix A). This test could be crucial for the confirmation of real seroconversion in pregnancies in women with negative or equivocal anti-*T**oxoplasma* IgG and positive anti-*Toxoplasma* IgM antibodies on the first samples, thus avoiding the wait until the IgG seroconversion.

The high number (27.9%) of equivocal Toxo II WB IgM results in the false IgM group, 19 of 68 patients (Table 2), shows the necessity of using two bands for defining the positivity of the test. This nonspecific reaction, which is most often a P30 alone, might be one of the sources of a false-positive result for the other IgM techniques in the samples.

Treatment (spiramycin) of the 56 patients from Pavia was started immediately after the detection of IgM in the absence of IgG, but no difference in the time to appearance of IgG was recorded compared with the other patients in centres where therapy was started only after documented seroconversion. LDBio Toxo II IgG and IgM WB allow an early and precise diagnosis of seroconversion, even in pregnant women who had been treated with spiramycin. All pregnant women who seroconverted can be treated, counselled and, if necessary, addressed to prenatal therapy and diagnosis earlier. Moreover, most women with a nonspecific response can be reassured, have their treatment stopped if it had been started and after another sampling 1 and 2 months after therapy interruption, they can be considered negative. With this study, we also confirmed the good sensitivity and specificity of LDBio Toxo II WB IgG, as well as its superiority in detecting seroconversion faster than Elisa, as reported in the literature. In about half of the patients (59.1%), samples tested positive earlier with WB compared with conventional techniques, as already reported in previous studies [8,10,12,13,16,19]. The possibility of investigating the immunodominant band confirmed the correct way to interpret the results of the test for IgM and made it possible to validate the specific profile: two bands between P30, 31, 33, 38 and 40 (including the P30), as defined by the manufacturer. Further studies and larger use of the method, if Toxo II IgM western blot is marketed, could confirm these data.

## Figures and Tables

**Figure 1 pathogens-11-00665-f001:**
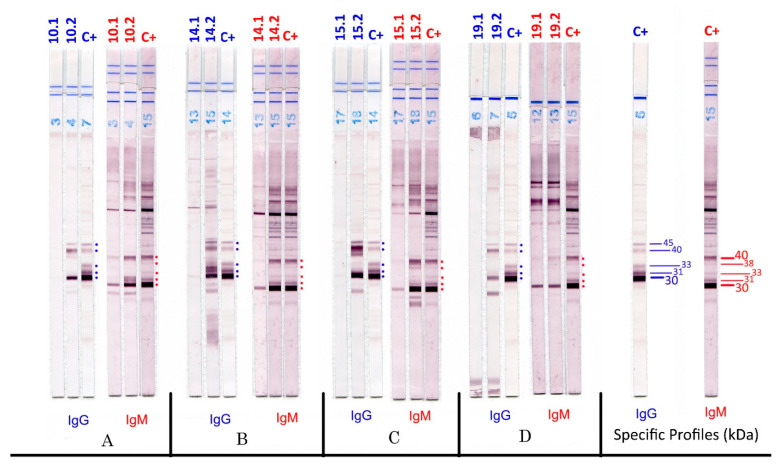
IgG and IgM Toxo II WB in four patients with seroconversion. (**A**–**D**): patient samples at Time 0, sample number 1 (negative IgG, positive IgM with traditional tests); 1 month later, sample number 2 (equivocal or positive IgG positive IgM with traditional tests); C+, positive control.

**Figure 2 pathogens-11-00665-f002:**
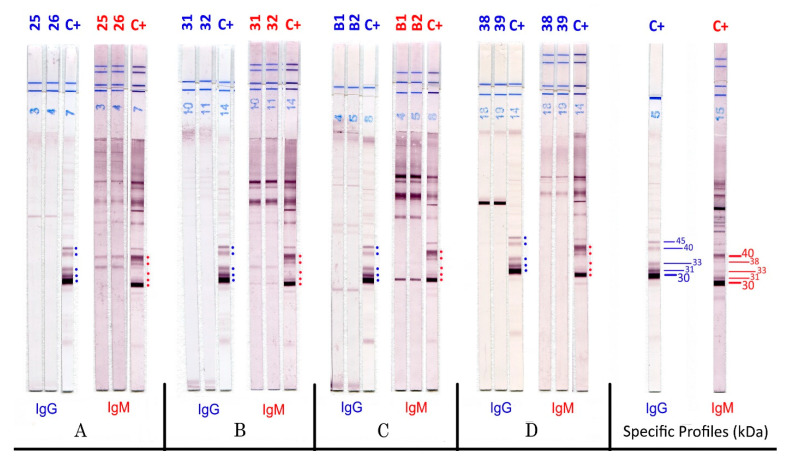
IgG and IgM TOXO II WB in four patients with false-positive IgM. (**A**–**D**), patient samples at different times, time 0 (left band for each patient and each isotype) and one month later (middle band for each patient and each isotype). All patients were negative IgG, positive IgM with traditional tests. C+, positive control.

**Table 1 pathogens-11-00665-t001:** Number of analysed samples per patient and test routinely used in each centre.

Centre	Seroconversions		Aspecific IgM		Tests
	Samples	Patients	Samples	Patients	
Grenoble	48	20	38	17	IgM ISAGA (bioMérieux, France) VIDAS TOXO IgG-IgM (bioMérieux, France), Toxo IgG-IgM Architect (Abbott Diagnostic, Germany)
Marseille	69	25	4	2	IgM ISAGA (bioMérieux, France) Toxo IgG-IgM Architect (Abbott Diagnostic, Germany)
Paris	45	20	60	21	IgM ISAGA (bioMérieux, France) LIAISON^®^ Toxo IgG, IgM, (Diasorin, Italy) Toxo IgG-IgM Platelia (Bio Rad, France)
Pavia	67	28	56	28	IgM ISAGA (bioMérieux, France) VIDAS Toxo IgG, VIDAS Toxo Avidity,) LIAISON^®^ Toxo IgG, IgM, (Diasorin, Italy), Home-made IGRA (Interferon gamma release assay)
Total	229	93	158	68	

**Table 2 pathogens-11-00665-t002:** Diagnostic accuracy of IgG and IgM WB for toxoplasmosis in 161 pregnant women.

Diagnosis	Toxo II IgM WB				Toxo II IgG WB		
	Positive	Equivocal	Negative	Total	Positive	Negative	Total
Seroconversion	91	2	0	93	92	1	93
False-positive/ nonspecific IgM	7	19 ^1^	42	68	0	68	68
Total	98	21	42	161	92	69	161
Sensitivity (equivocal considered negative)	97.8% (95CI 91.7–99.6%)				98.9% (95CI 93.3–99.9%)		
Sensitivity (equivocal considered positive)	100% (95CI 95.0–100%)						
Specificity (equivocal considered negative)	89.7% (95CI 79.3–95.4%)				100% (95CI 93.3–100%)		
Specificity (equivocal considered positive)	61.8% (95CI 49.1–73.1%)						

Results obtained according to diagnosis, sensitivity and specificity of the Toxo II WB IgG and IgM in a selected population: 95CI, 95% confidence intervals. ^1^ One patient had both equivocal and positive results and was, therefore, classified in the positive group.

## Data Availability

All the data are available in the lab of Valeria Meroni.

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
