# Peer review of "Diagnostic Accuracy of LDBIO-Toxo II IgG and IgM Western Blot in Suspected Seroconversion in Pregnancy: A Multicentre Study"

_pathogens, 2022, doi:10.3390/pathogens11060665_

Round 1

Reviewer 1 Report

This manuscript reports the results of the authors’ interrogation of the clinical utility and diagnostic accuracy of two commercial western blot kits, Toxo II IgG and Toxo II IgM (LDbio Diagnostics), in the pursuit of expedited diagnosis of Toxoplasma seroconversion in pregnant women. The authors are particularly interested in establishing a means for earlier diagnosis of seroconversion in order to expedite therapeutic interventions to prevent adverse consequences of congenital toxoplasmosis. The authors present a clear and appropriate approach.  Some problems with the manuscript arise in the presentation of the results and the conclusions.  Overall the authors are to be congratulated on a well-planned and written manuscript.  

The problem stated, and the objective are clinically quite important; the overall experimental approach seems straightforward. This manuscript has mainly minor shortcomings some of which lead to confusion.

Materials and Methods 

Line 82: “was very low, often negative”.  This is too vague. Realizing that many different traditional reference tests were used, it would be helpful to the reader to add a simple supplemental table outlining the numerical ranges for “positive”, “negative”, and “aspecific IgM” for each commercial test.

Methods

Line 119:  What is meant by “The bands used for IgG and IgM WB are close to each other…”?  This implies that p30 in the IgG test is not the same as p30 in the IgM test.  Is this the intended meaning?

Line 121: “the first evaluation” suggests that the test was part of the current investigation.  This is probably not what is meant.  Please re-word this.

Statistical Analysis

Line 136: “do to repeated samples” should be “due to repeated samples”

Results

Lines 160 – 164:  This section is somewhat confusing.  What are “the discordant results”. What is meant by the claim that the appearance of IgM or the appearance of IgG in seroconversion “was recorded by WB before the traditional tests”.  Does this imply the tests were done on serial samples from the same patient?  How much time elapsed between each sample?  Is it days, weeks, or …?  

Lines 164 – 167.  What are the proteins represented by P30 and P40? It would be more interesting and informative to the reader for the authors to add a little information about these proteins.

Figure 1:  The legend refers to Time 0 and 1 month later.  But the strips are not labeled with these designations.  Please re-label or add the designation “Time 0 (sample number .1) and 1 month later (sample number .2)”, or equivalent, so that it is obvious.

Figure 2:  Same labeling problem as in Figure 1 but here it is a completely different sample number.  Please make the labeling consistent and clear.

Conclusions

Line 197:  The paragraph is referring to the IgM WB, but isn’t the S2 file about the IgG WB?

Lines 201 – 203:  It is not clear from where the 29.4% number came.  Further, was there a mention of a nonspecific reaction to only P30?

Line 206:   What is meant by “no difference was recorded”?

Lines 215 – 217:  Is this sentence about results reported in the literature or does it also refer to data presented in the current manuscript?

Lines 217 – 220:  Due to the small number of samples, this statement might be premature.  They might be compelling data that warrant more testing.

Supplementary Files 1 and 2: Correct frequent use of “sensibility” to “sensitivity”

Reviewer 2 Report

Well conducted study by experts in the field. The authors show the interest of using LDBio western-blot tests for the diagnosis of seroconversions and the differentiation between specific and non-specific IgM. The tests used are the WB Toxo II IgG test which is already widely available and whose performance is known, and an as yet unmarketed LDBio Toxo II IgM test.

A few remarks:

  • The fact that Toxo II IgM is a new, non-commercialized test must appear more clearly in the abstract and the introduction, as the evaluation of this new test seems to me to be the main objective of this study, even more than the combined use of the 2 western-blot tests.
  • In the supplementary data, in the comparison with Biorad IgM and with ISAGA, it appears that several sera remained constantly negative with WB Toxo II IgM. This may be presented and discussed in the discussion part of the main paper. Does this correspond to seroconversion without detectable IgM with WB Toxo IgM?
  • In the supplementary data concerning comparison for IgG, tables are confusing: for the cases “False positive” which are indicated in the text above the tables as negative samples, it would be clearer to specify “False positive IgM”.

Minor points:

Supplementary data: the word “sensibility” is used several times instead of “sensitivity”.

Introduction, L.53: “treatment with spiramycin, …, can delay IgG production …”: isn’t also the case (and probably even more) for treatment with sulfadiazine plus pyrimethamine which is the alternative treatment?

L.136: “…analysis at performances do to repeated samples of the same patient.”: “due to” instead of “do to” ?

This work was funded by LDbio, but I didn't find any declaration for links or conflict of interest. Shouldn't this appear?
